# Gender Differences in Adventure Tourists Who Practice Kayaking in Extremadura

**DOI:** 10.3390/ijerph20053889

**Published:** 2023-02-22

**Authors:** Jorge Rojo-Ramos, Santiago Gómez-Paniagua, Juan Carlos Guevara-Pérez, Jorge García-Unanue

**Affiliations:** 1Physical Activity for Education, Performance and Health (PAEPH) Research Group, Faculty of Sports Sciences, University of Extremadura, 10003 Cáceres, Spain; 2BioẼrgon Research Group, University of Extremadura, 10003 Cáceres, Spain; 3Faculty of Economics and Business, University of Zaragoza, 50005 Zaragoza, Spain; 4IGOID Research Group, University of Castilla-La Mancha, 45004 Toledo, Spain

**Keywords:** adventure sport, kayaking, economic impact, gender, nature

## Abstract

Adventure tourism is among the sectors that has experienced the greatest growth in participation in recent years. In addition, it provides a unique opportunity to generate different benefits for rural populations as well as for the preservation of their environment. The objective of this study was to analyze gender differences in the profile, estimated expenditure, perception of economic impact and satisfaction of adventure tourists visiting the Valle del Jerte (Extremadura, Spain) for kayaking activities. The sample was composed of 511 tourists who kayaked in the Valle del Jerte. Gender differences were analyzed by the Mann-Whitney U test in continuous variables and Pearson’s chi-square test in categorical variables. The predominant profile of the kayaking tourist is Spanish, married, employed, with university education, lives with their partner with children at home, chooses a rural home for accommodation, travels with companions, uses their own car for transportation, spends 550 euros on average, has good perceptions of the economic impact of the activity on the destination and is satisfied with the kayak service received. This information is relevant for public and private organizations as well as for the local community to be able to offer services more oriented to the tourist who engages in these activities, as well as to attract more tourists.

## 1. Introduction

Over the last few decades there has been a large increase in the number of people demanding sporting activities in the natural environment, becoming as one of the most important economic and social development strategies for rural areas [1]. In this context, adventure tourism is one of the sectors that generates the most economic benefits and attracts the most participants [2], offering thrilling commercial guided tours that involve outdoor activities based on local geography [3], kayaking being one of the most practiced modalities in this field [4]. Adventure tourism’s rise has been ascribed to its advantages for enhancing subjective well-being and promoting good mental health [5]. Adventure seekers look forward to interactions and experiences that are stimulating, physically demanding, or otherwise affect their subjective well-being and satisfaction [6]. Likewise, the popularity of tourism and aquatic sports is rising everywhere in the world, with large bodies of water being used for a wide variety of outdoor recreational activities [7]. Although river regulatory projects and water activities involving scooters and motorboats limit other aquatic activities because they are fully associated with the available natural resources, river and sea kayaking is a growing subset of adventure tourism [8]. In addition, the COVID-19 pandemic restricted the possibility of access to various tourism activities, transforming the sector and favoring local tourism over global tourism [9]. 

Adventure tourism encompasses activities including skydiving, mountain climbing, surfing, rafting, snowboarding and kayaking [10]. Kayaking is defined as an adventure tourism modality that includes the navigation of rivers by use of kayaks, with no other means of propulsion, and boat propulsion and control of the boat other than that generated by the navigators themselves with the use of paddles [11]. Kayaking has also been one of the most explored activities in the scientific literature [12], proving to be one of the major attractions for certain tourist locations [13]. In adventure tourism, gender has been pointed out as one of the main determinants of participation and specialization [14], men being more likely than women to get involved in and participate in outdoor recreational activities on a regular basis [15]. According to O’Connell’s [16] kayaking research, women were more highly motivated than men by an appreciation of nature and creativity, while kayaking is a predominantly male activity in which gender differences are more visible than in other adventure activities [17]. 

Taking all the above into account, the aim of this study was to explore gender differences in the economic impact, perceptions and satisfaction among adventure tourists who visited Valle del Jerte region (Spain) to carry out kayaking activities. In this way, the whole environment that is part of the tourist destination, companies and the local population, have relevant information to develop different strategies to encourage the participation of adventure activity participants, as well as the adaptation of their services to the profile of the tourists who are likely to hire them. Furthermore, to the authors’ knowledge, this is the first approach that explores gender differences in the context of adventure tourism in Spain.

## 2. Literature Review

### 2.1. Socioeconomic Impact

Regarding adventure tourism, the sector’s expansion and the rise in leisure activities would have a substantial impact on local economies, especially in terms of the socioeconomic advancement of peripheral areas and rural locations [18], contributing to the maintenance of regional identity and for the promotion of local service activities, which, in turn, helps to create jobs [19,20], as well as for the modernization and development of these areas [21]. To this end, public-private partnerships and the effective implementation of local development initiatives, plans, and strategies are necessary [22]. Furthermore, locals ought to be able to coordinate the resources and themselves for the benefit of everybody in a way that allows local communities to manage their own resources and utilize them for economic gain [23]. In this sense, and according to Manente, Minghetti, and Mingotto [24], tourist planning aims to build methods for delivering social-economic advantages for society to achieve sustainable development of tourism by preserving the local environment and culture. All the elements of tourism, including tourist sites and activities, transportation, infrastructure, and institutional framework, should be taken into account while planning [25].

In this context, methods are needed to classify tourists to predict their potential impact. Expenditure-based tourist segmentation has been recognized as having unrealized practical potential for the tourism industry and tourist destinations, because its analysis is characterized by a very generic analysis of the tourist profile [26]. In this line, it is well known that tourists who visit the same place at the same time use their budgets in quite diverse ways [27]. In addition, the same amount spent on various tourist services has a distinct influence on the local economy [28]; therefore, destinations ought to pay closer attention to visitor market segments with particular spending habits [29]. Despite all this, most scientific work has assessed tourists according to their total expenditure, classifying them into high, medium and low levels of expenditure [30,31]. By contrast, trip budget allocation refers to the percentage of expenses that is dedicated to accommodation, diets, transportation, and other related issues [32], providing more in-depth economic modelling of tourism expenditure and analysing issues more closely related to how tourists spend [33]. However, adventure tourism, and especially kayaking activities, do not seem to follow the same trend [34], mainly due to their diverse demographic profiles and travel behaviors [35].

### 2.2. Tourist Perceptions and Satisfaction

Tourism research has paid a lot of attention to how locals perceive the impact of tourism, and support its growth within a place [36], to better understand how tourism can be most advantageous for all parties involved [37]. Nevertheless, this work does not take into account the possibility that visitors could develop an emotional bond with their travel destination and occasionally identify with it [38,39]. As a result, travelers may become conscious of how their actions affect the environment and culture [40]. Destinations chosen by visitors are a result of what they experience and observe [41]; for instance, if most of the effects of tourism in a place are unfavorable, visitors are less likely to return, as tourists’ attitudes are the main factor guiding their behavior [42]. Thus, asking tourists about how they think tourism affects the location, and what contributes to more favorable opinions about tourism development, would be both rational and valuable [43].

Another factor to be taken into account in destination selection and loyalty is customer satisfaction with service, as customer ratings of products reflect the potential success of the business [44]. The extent to which an experience elicits favorable emotions is referred to as satisfaction [45]. Customer satisfaction has historically been seen as a crucial business objective because it is believed that satisfied customers purchase more [46] and are more likely to have a strong sense of loyalty to a place [47]. Some experts assess the final level of satisfaction by comparing initial expectations and perceived performance following consumption [48]; in other words, a consumer is satisfied if they feel they have received more value than they anticipated. 

### 2.3. Gender Differences

Tourist-related gender studies have helped us comprehend tourism management from a gender-based perspective [49]. Gender is one of the most important factors for successful marketing and management, as evidenced by the growing body of research on gender-related marketing and management strategies [50]. Therefore, a lot of scholars have studies gender and its significance in tourism studies, frequently highlighting gendered experiences, variations in men’s and women’s vacation preferences, and gender disparities in tourist development [51]. However, research on gender issues in the adventure tourism sector continues to be very scarce and often with very disparate results. According to an early study [52], the majority of American adventure tourists are men (68%), who frequently prefer hard adventure, whereas women are more likely to partake in soft adventure. On the contrary, more recent studies [2] point to a change in the trend, where men represent 57% of the participants with no major differences in adventure modalities (hard vs. soft), as well as an increase in the number of female adventure tourists traveling alone [35]. Likewise, there may be differences in preferences for adventure activities when gender and age of the person are related, with women preferring more age-related activities such as bird watching and hiking, while older men may prefer activities that are more often associated with younger people, such as climbing, caving and rafting [53]. 

In the motivational context, men and women participate in different forms of outdoor adventure activities at different rates and to different degrees of specialization, men being more likely than women to participate in outdoor recreation activities on a regular basis [54]. While male participation in leisure activities is largely motivated by competition and challenge [14], females are more driven by the desire to develop new skills in a supportive outdoor environment, break free from traditional gender roles, and make connections with other women [55]. According to Ewert et al. [56], female rock climbers, white-water kayakers, sea kayakers, and canoeists placed a higher value on social objectives, such as making friends and belonging to a team, while men were more driven by self-image and sensation-seeking than women. In addition, research on sea kayakers ascertained that females were more highly motivated by the enjoyment of nature and creativity than males [16]. 

## 3. Materials and Methods

### 3.1. Participants

The sample consisted of 511 tourists who kayaked in the Valle del Jerte accompanied by professional guides through services offered by local active tourism companies. Participants were selected using simple random sampling. Of the total sample, 206 (40.3%) were men and 305 (59.7%) were women.

### 3.2. Procedure

To access the sample, active tourism companies in the Valle del Jerte (Extremadura, Spain) were asked to collaborate. Those that agreed to participate in the study were visited by a member of the research team to inform them of the procedure and method of sample selection. The procedure developed was as follows: each group of participants who had done the kayaking activity with the collaborating companies and who decided to collaborate in the study, would participate in a sweepstakes whereby users would have to draw a piece of paper from a drum. The user on whose paper an X symbol was drawn was asked to take the survey once he/she had accepted the informed consent. To access the survey they were given a QR code they had to scan with their mobile device, and they would go directly to a form created with the Google Forms tool. As a reward, all group members were given a reusable metal bottle donated by the SOPRODEVAJE Rural Development Group for collaboration in the study. The average time to complete the form was 12 min. All data were collected and processed anonymously.

It was decided to use an e-questionnaire because of all the advantages it offers [57,58] since all the responses were collected in the same database automatically regardless of the company with which the survey was conducted, and it also facilitated a higher response rate since it was considered easy for participants to access the URL through a QR code. Data were collected between the months of March to October 2021 and March to October 2022.

As inclusion criteria, participants had to be of legal age, have done the kayaking activity through a collaborating active tourism company, and have a cell phone to access the questionnaire. 

### 3.3. Instruments

A questionnaire with 10 sociodemographic questions (sex, age, height, weight, nationality, autonomous community of residence, marital status, professional situation, level of studies and type of household of residence), and six questions related to the profile of the kayaker in the Valle del Jerte (type of accommodation chosen for the trip, accompanying athletes, means of transport used to the Valle del Jerte, trip planning based on advertising, number of accompanying persons and number of overnight stays) was prepared in order to characterize the sample.

Expenditure was also estimated using two scales. The first scale was composed of five questions regarding the estimated amount they would spend or had spent on the trip, on lodging, transportation, kayaking, breakfast, lunch and dinner. The second scale was composed of an initial question “Could you specify how much you have spent approximately in the Valle del Jerte in the following concepts?”, in which the user had to report from a scale composed of the following ranges “from 0 to 14 euros”, “from 15 to 29 euros”, “from 30 to 44 euros”, “from 45 to 59 euros”, “from 60 to 74 euros”, “from 75 to 89 euros”, “from 90 to 104 euros”, the money spent on press, active tourism sports activities, bars and restaurants, souvenirs, museums/visits, sports equipment or others.

Later, a questionnaire on the perception of the impact of kayaking in the Valle del Jerte was administered, consisting of a Likert-type scale (1–5) with 1 totally disagreeing, 2 disagreeing, 3 indifferent, 4 agreeing, 5 totally agreeing. The scale evaluated the opinion of the participants on the impact of kayaking in the Valle del Jerte, and was composed of nine items.

Finally, a questionnaire was used to evaluate satisfaction with the activity and the service received. This questionnaire consisted of two questions related to the user’s satisfaction with the service provided by the adventure sports and active tourism company with which he/she had practiced kayaking (from 1 to 10) and whether he/she would return to the Valle del Jerte to do the same adventure sports activity or another one.

### 3.4. Statistical Analysis

First, the Kolmogorov-Smirnov test was used to explore the distribution of the data and observe whether the assumption of normality was met. This assumption was not met, so it was decided to use nonparametric statistical tests. Pearson’s chi-square test was used to analyze the differences between the item “Would you return to the Valle del Jerte to do the same or other sports activities in nature?” and the variables included in the sociodemographic questionnaire and profile of the tourist according to gender. The Mann-Whitney U test was used to analyze the differences between the item “Has satisfaction with the service provided (kayaking) been?” and the variables of the questionnaire of estimated expenditure and perception of the impact of kayaking in the Valle del Jerte as a function of sex, as well as in the last two variables of the sociodemographic and tourist profile questionnaires. Continuous variables are presented as mean and standard deviation, and categorical variables are presented as number and percentages. The significance level was set at *p* < 0.05.

## 4. Results

Table 1 shows the sociodemographic characterization of the sample according to gender. 

Statistically significant differences were found according to gender in the variable “professional situation”. The profile of the kayaking tourist is a Spanish woman, from the Autonomous Community of Extremadura, married, employed with a permanent contract, with university education, living with his partner with children at home, with a mean age of 35.47 years and a mean BMI of 26.86.

Table 2 presents the descriptions of the variables used to evaluate the type of accommodation chosen for the trip, the type and number of companions, the means of transport used to travel to the Valle del Jerte, the advertising that influenced the choice of destination, and the number of overnight stays. 

The profile of the kayaker is one who chooses a rural home for an average of 2.32 nights, usually being accompanied by family members (accompanied by an average of 2.62 people) and the means of transport chosen to travel to the destination is their own car or a rental car. The kayakers do not usually plan the trip through any advertising; however, the main means of advertising is a channel that comes through family or friends. Statistically significant differences were found in the variable “type of accommodation”, with women being more likely not to spend the night at the destination or to choose a hotel, motorhome or rural house for the night. By contrast, men were more likely to stay at a campsite than women. Similarly, statistical significance was obtained when evaluating the number of accompanying persons with men showing higher values.

Table 3 presents the descriptives and differences of the first expenditure estimation scale. Women estimated spending more money than men at a general level throughout the entire trip. Statistically significant sex differences were found in the items “how much did you spend in euros on transportation (round trip)?” and “how much have you spent on kayaking activities in the Valle del Jerte?” with men spending the most money on these items.

Table 4 shows the descriptives and frequencies of the second expenditure estimation scale. Male kayakers spent more than females on the items or services listed. Statistically significant differences were found according to sex in the variables “active tourism sports activities”, “bars and restaurants” and “museum/visits”.

Table 5 shows the descriptives and differences of the scale designed to evaluate the perception of the tourist regarding the impact of kayaking in the Valle del Jerte. There was a predominant perception in both sexes that kayaking represents an element of pride and satisfaction for the Valle del Jerte. Men only scored higher than women on the variables “increased recognition and promotion of the destination” and “public spending is necessary for the development of this type of sports”. In the remaining items, women scored higher than men. However, statistically significant differences by gender were found in the items “generates a traffic and safety problem”, “disrupts the usual rhythm and impairs other activities”, “employment increases” and “economic advantages are concentrated in the hands of a few companies and individuals”.

Finally, we evaluated the satisfaction of the tourist who had kayaked with a local company in the Valle del Jerte (see Table 6). Almost the entire sample would return to the Valle del Jerte to do the same or other sports activities in nature, and were very satisfied with the service provided (Mfemale = 9.91 and Mmale = 9.89). No statistically significant differences were found in reference to gender in any of the variables.

## 5. Discussion

This research originated from the need to know the gender differences present in adventure tourism, and was motivated mainly by the growing popularity of adventure tourism, including visits the Valle del Jerte to practice kayaking activities. It was possible to assess in depth both the profile of the tourist and the estimation of their spending, their perceptions about the impact generated by these activities in the local community and their satisfaction with the contracted sports service. 

A study carried out in South Africa showed that the majority of kayakers are single men who have completed only secondary education [59], contrary to the present study. Similarly, Albayrak and Caber [60] analyzed the demographic profile of German tourists visiting Turkey for rafting activities, with most of them being single males, employed in private companies, undergraduate students, and with no previous rafting experience. Hasan and Neela [61], on the other hand, indicated that sports tourists practicing adventure water activities in Malaysia were predominantly male, with an age range between 18 and 34 years old, university graduates and service sector workers. In addition, in Norwegian domestic tourism in the nature activities sector, the predominant demographic profile included both genders, an age between 35 and 44 years old, married, with a high school education and no children. In the Spanish context and more specifically in Extremadura, adventure tourists tend to be Spanish, single, university educated, with permanent employment contracts, with children at home and with a median age of 40 [62]. As for gender, previous research had already pointed out possible distribution differences in terms of professional status, marital status and educational level among adventure tourists who had visited Malaysia [63]. Similarly, Rojo-Ramos et al. found significant differences in the level of education and professional status when assessing gender in adventure tourists [62]. It is worth mentioning the scarce bibliography with a segmentation of the different tourism sectors, in this case adventure tourism, in which the dependent variable is gender.

As for the tourist profile, significant differences were found in the type of accommodation and the number of accompanying persons when evaluating the gender variable in the sample of sports tourists. Valek et al. [64] found that sports tourists traveling within their own country, as in this study, chose the most economical options for accommodations. Previous studies showed that the preferences in the choice of accommodation during vacations varied according to the gender of the tourist [65], as well as the presence or absence of children on the trip, and their age [66]. Similarly, Ariffin and Maghzi [67] found modifications in the satisfaction of tourists staying in hotels depending on the sex of the tourist, whereby men might choose other types of accommodation due to their higher expectations. Omar and collaborators showed that women give more importance to accommodation services when choosing a destination than men do [68], although a study of kayaking tourists in Poland showed that men place greater importance on accommodation than women [69]. Other research focusing on Spanish sports tourists found differences in distributions according to the type of accommodation chosen when analyzing gender [70]. By contrast, it has been shown that gender differences do not exist when it comes to the choice of shared accommodation [71]. Chen et al. [72] suggest that traveling with a companion is an important factor when engaging in tourism activities in nature. Marujo and colleagues study [73] reported that 82.2% of those surveyed traveled with companions, 26% of whom were organized groups and 25% their partner. Albayrak [74] found that the majority of Turkish adventure sports tourists traveled mainly with family members. The most common tendency for adventure tourists is to travel with friends who demand the same type of activities [75,76]. In this context, women may be greatly discouraged from participating in certain sports activities due to a lack of friends or because they do not know anyone [77]. Similarly, Buckley and Westaway [78] support the idea that women demand a greater need for companions in adventure tourism than men because they pay more attention to safety issues [79]. 

One of the most common ways of segmenting the market to analyze tourism expenditure is by gender [80]. This is created by different genders wanting different products/services from their purchasing decision [81], with men looking for more adventure experiences, so their spending increases on these activities. Sato et al. [82] noted that, in general, men spend more money on sports tourism-related activities than women. This idea is supported by a study conducted on participants of Trail events in Croatia and Slovenia, where women had lower expenditure than men, but found no differences in the Mountain Bike modality [83]. Similarly, a study carried out in Norwegian mountain areas showed that female tourists paid more attention to expenditure than men, even though they carried out the same activities [84]. Fredman [85] found that male visitors spend more money on activities or services outside of the sporting activity for which they have traveled to the mountain. On the contrary, numerous studies have shown a completely opposite trend. Wicker and colleagues [86] showed that gender had a significant positive influence on daily expenditure, indicating that women spent more money than men as tourists attending marathon events. Xue and Zhang [87] observed that women spent more on hotel services, because they demanded higher quality, but found no differences in expenses related to restaurants. Cheung [88] found that women spend more than men on purchases, but there were no differences in transportation, accommodations or food. English tourists who took a single day trip to participate in outdoor sports activities showed no differences in total spending by gender [89]. Likewise, tourists who followed the American golf circuit show no differences according to gender in relation to expenditure, as well as golfers who regularly practice this activity in Germany [90]. This finding is also supported by Dixon et al. [91], who found no inequalities in the daily expenditure of mountain bike riders. Finally, those tourists who practice winter sports showed similar total spending regardless of gender [92].

Regarding the impact of adventure activities, according to Steynberg and Grundling [93], this correlated with the amount of activities present in a given location because these activities occur in places that are not suitable for mass tourism due to their isolation and distinctive natural or cultural heritage. According to Muresan et al. [94], the growth of rural tourism has the potential to enhance community wellbeing due to its favorable impact on income production, developing strong connections with industries producing food and agriculture to create synergies that will lead to sustainable development in the long run [95]. As indicated by Duglio and Beltramo [96], participants’ perceptions of the social and economic impact of small-scale adventure activities are very positive. In the same way, participants and attendees of a sporting event clearly perceive that it increases the recognition and promotion of the destination, representing an element of pride and satisfaction for the destination [97]. This idea is also supported by the participants of an event in Hong Kong who claim that the city’s image on an international level would be improved [88]. The impact of sports activities on job creation has also been studied, both in sporting events and in adventure activities related to active tourists [98,99], with sports businesses, food and lodging services being the most impacted. As in this study, participants in adventure activities in a natural park in the United States did not perceive that these activities had a negative impact on traffic or safety, nor impede the development of other tourist and sporting activities [100]. Another study carried out in Thailand explored the perceptions of tourists involved in activities in the natural environment, where tourists perceived taking part in different services provided by the local community as a whole with clear indications on how to generate the minimum possible environmental impact [101]. 

Given that most participants in this type of activities want intense experiential results, aquatic activities provide a suitable environment for the study of adventurous experiences [102]. The pursuit of high visitor satisfaction, particularly for outdoor recreation activities, has been viewed as a crucial goal for administrators offering such leisure services [103], because this satisfaction has been highly correlated with the tourist’s intention to revisit [104]. In this sense, Ponte et al. [105] reported good satisfaction values in tourists who traveled to the Azores (Portugal) to carry out adventure activities, with 71% of positive responses when considering the option of revisiting the destination. Equally, aquatic activities developed for tourism in India showed both high expectations and high satisfaction [106]. However, 35% of customers rated their satisfaction with an adventure tour in Peru as high, while 50% of customers rate it as fair [107]. Similarly, tourists who came to Germany for the purpose of mountain activities experienced a very low level of satisfaction, mainly due to fear [108]. Regarding gender differences, Rohman [109] found that the levels of satisfaction experienced by adventure tourists in Indonesia did not differ when analyzing the gender of the participants. Additionally, Amatulli et al. [110] found no differences in the satisfaction generated by mountain biking activities in Italy when assessing gender. Gender did not seem to have an effect on the satisfaction of tourists traveling to Malaysia [111]. By contrast, Kumar [112] defined gender as a mediator of the levels of satisfaction expressed by rural tourists in India. 

### Practical Implications

First, tourism companies should consider the profile and preferences of the tourists that usually come to the location for kayaking activities, so that related services such as accommodation, transportation, advertising, type of nature practice or budget are taken into account when developing specific products and services. In addition, adventure tourism companies need to recognize gender equality and women’s voices by marketing the elements of their products that help women overcome their main barriers, as this aspect is still prevalent in most tourism destinations [77]. Therefore, all the issues discussed in this paper should be considered to facilitate the participation of both men and women in these activities. Similarly, tourism businesses and environments play a critical role in welcoming tourists at a psychological level, as the tourists’ experiences and perceptions are likely to deteriorate if they feel less welcome or are emotionally separated from residents and observe negative outcomes from their visits. However, it should be mentioned that the majority of the sample had high satisfaction and intentions to return to the destination. Finally, tourism companies should develop lines of study and research focused on the tourists themselves, their experiences, and their relationship with the destination environment, as these types of approaches provide valuable information to both public and private entities to develop different regional areas with a huge potential. Finally, future research in the field of adventure tourism can use this global gender analysis in the Spanish context as a reference.

## 6. Conclusions

This study allows us to characterize and evaluate the gender differences present in the profile, estimated expenditure, perceived impact and satisfaction of adventure tourists visiting the Valle del Jerte for kayaking activities. Analyzing the variables included in the study, the private or public organizations and the local population have a reference to adapt the services offered that include the realization of this adventure activity. 

Gender differences were found in most of the analyzed variables pertaining to tourists, such as sociodemographic characteristics, tourist profile, spending estimates and perceptions of their impact on the tourist’s life and the impact of the tourism industry. 

### Limitations and Future Research

This study has several limitations. First, the sample only included domestic resident tourism and did not analyze data on tourists who may arrive from abroad. If such data were included, the results could fluctuate and the measures proposed might not be consistent. Another important limitation is the selection of the sample according to the adventure activity developed. Finally, the scales that analyze the socioeconomic impact (expenditure and perceptions) have been adapted from several scientific studies in Spain but have not been previously validated in the context of adventure tourism.

As future lines of research, extending this analysis to all the adventure activities that take place in the Valle del Jerte would allow us to obtain a more global vision of the different issues that influence the behavior of tourists, as well as extending this analysis to the entire Autonomous Community of Extremadura. In the same way, it would be interesting to relate the different socio-demographic characteristics of the participating sample with the estimation of expenditure, perceptions of impact and satisfaction to carry out a deeper segmentation and assessment of the market. Inspecting the psychometric properties and validity of the scales used in the study would also be a major advance in this context.

## Figures and Tables

**Table 1 ijerph-20-03889-t001:** Characterization of the sample (N = 673).

Variable	Categories		N	%
Gender	Male		206	40.3
	Female		305	59.7
**Variable**	**Frequencies**			** *p* **
	**Total** **N (%)**	**Male** **N (%)**	**Female** **N (%)**	
**Nationality**				
Spanish	510 (99.8)	205 (99.5)	305 (100)	0.223
Foreign	1 (0.2)	1 (0.5)	-	
**Autonomous Community**				
Extremadura	450 (88.1)	177 (85.9)	273 (89.5)	0.220
Other Autonomous Community or Foreign	61 (11.9)	29 (14.1)	32 (10.5)	
**Marital Status**				
Single	192 (37.6)	65 (31.6)	127 (41.6)	0.085
Married	302 (59.1)	133 (64.6)	169 (55.4)	
Widowed	-	-	-	
Separate	16 (3.1)	7 (3.4)	9 (3)	
Divorced	1 (0.2)	1 (0.5)	-	
**Professional Status**				
Employer, professional or self-employed person who employs others	70(13.7)	40 (19.4)	30 (9.8)	0.002 *
Employer, professional or self-employed person who does not employ others	33 (6.5)	18 (8.7)	15 (4.9)	
Employee or employee with a permanent contract	195 (38.2)	77 (37.4)	118 (38.7)	
Employee or employee with a temporary contract	126 (24.7)	46 (22.3)	80 (26.2)	
Unemployed	87 (17)	25 (12.1)	62 (20.3)	
**Level of Studies**				
Primary education or less	-	-	-	0.362
Secondary education, first stage	6 (1.2)	2 (1)	4 (1.3)	
Secondary education, second stage	113 (22.1)	52 (25.2)	61 (20)	
University education	392 (76.7)	152 (73.8)	240 (78.7)	
**Type of Household**				
Single parent cohabitation with a child	17 (3.3)	11 (5.3)	6 (2)	0.059
Single parent cohabitation with more than one child	26 (5.1)	14 (6.8)	12 (3.9)	
Couple without children cohabitation at home	158 (30.9)	57 (27.7)	101 (33.1)	
Couple with children cohabitation at home	186 (36.4)	82 (39.8)	104 (34.1)	
Single with no children	65 (12.7)	25 (12.1)	40 (13.1)	
Lives with a relative	58 (11.4)	17 (8.3)	41 (13.4)	
Shared household	1 (0.2)	-	1 (0.3)	
**Variable**	**M (SD)**	**M (SD)**	**M (SD)**	** *p* **
Age	35.47 (7.78)	35.37 (7.71)	35.53 (7.84)	0.724
BMI	26.86 (3.64)	27 (3.44)	26.76 (3.77)	0.604

Chi-Square and Mann-Whitney U test *p*-values * *p* is significant <0.05. M = mean value; SD = standard deviation; N = number; % = percentage.

**Table 2 ijerph-20-03889-t002:** Tourist profile.

Variable	Frequencies			*p*
	Total N (%)	Men N (%)	Women N (%)	
**Type of accommodation**				
I will not stay overnight	83 (16.3)	31 (15)	52 (17.1)	<0.001 *
Hotel or aparthotel	66 (12.9)	28 (13.6)	38 (12.5)	
Caravan or camper parked on public roads	5 (1)	-	5 (1.6)	
Shelter	1 (0.2)	-	1 (0.3)	
Full housing for rent	9 (1.8)	4 (1.9)	5 (1.6)	
Room in a private house	2 (0.4)	-	2 (0.7)	
Rural home	188 (36.9)	58 (28.2)	130 (42.8)	
Camping	154 (30.2)	85 (41.3)	69 (22.7)	
Housing property	-	-	-	
Friend’s or company’s family home	-	-	-	
Home exchange	-	-	-	
Other accommodations	2 (0.4)	-	2 (0.7)	
**I have come to practice kayaking accompanied by**				
Family	437 (85.5)	182 (88.3)	255 (83.6)	0.135
Friends	74 (14.5)	24 (11.7)	50 (16.4)	
Alone	-	-	-	
**Means of transport used to travel to the Valle del Jerte**				
Own or rented car	315 (61.6)	123 (59.7)	192 (63)	0.460
Carpooling with payment to the driver	196 (38.4)	83 (40.3)	113 (37)	
Bus	-	-	-	
Train	-	-	-	
Non-motorized land transportation	-	-	-	
**Have you planned your kayaking trip to the Valle del Jerte as a result of any publicity?**				
No	284 (55.6)	100 (48.5)	184 (60.3)	0.050
Press	1 (0.2)	-	-	
Radio	-	-	-	
Television	3 (0.6)	-	3 (1)	
Social media	30 (5.9)	14 (6.8)	16 (5.2)	
Friends/family	167 (32.7)	78 (37.9)	89 (29.2)	
Others	26 (5.1)	14 (6.8)	12 (3.9)	
**Variable**	**M (SD)**	**M (SD)**	**M (SD)**	** *p* **
How many people have you come to practice kayaking with (not including yourself?)?	2.62 (1.32)	2.73 (1.15)	2.55 (1.42)	0.005 *
How many nights will you spend in the Valle del Jerte?	2.32 (0.92)	2.46 (1.08)	2.22 (0.78)	0.057

Chi-Square and Mann-Whitney U test *p*-values * *p* is significant <0.05. M = mean value; SD = standard deviation; N = number; % = percentage.

**Table 3 ijerph-20-03889-t003:** First expenditure estimation scale.

Variable	Gender		*p*
	Female M (SD)	Male M (SD)	
How much do you expect to spend (in euros) in the Valle del Jerte during your stay?	554.34 (207.15)	542.42 (262.64)	0.805
Could you please specify how much you have spent (in euros) approximately on accommodation including your companions?	150.18 (151.80)	171.58 (155.01)	0.123
How much did you spend in euros on transportation (round trip)?	45.39 (36.74)	52.77 (43.70)	0.016 *
How much have you spent on kayaking activities in the Valle del Jerte?	43.18 (87.45)	47.24 (81.99)	0.003 *
How much do you expect to spend (in euros) on breakfast/lunch/dinner during your stay?	126.07 (75.85)	137 (87.50)	0.193

Mann-Whitney U test *p*-values. * *p* is significant <0.05. M = mean value; SD = standard deviation.

**Table 4 ijerph-20-03889-t004:** Second expenditure estimation scale.

Could you Please Specify Approximately How Much Has Been Spent in the Valle del Jerte on the Following Items?	Gender		*p*
	Female M (SD)	Male M (SD)	
Press	1.04 (0.23)	1.03 (0.25)	0.887
Active tourism sports activities	2.49 (1.65)	3 (1.68)	0.006 *
Bars and restaurants	2.46 (1.06)	3.49 (1.91)	<0.001 *
Souvenirs	1.15 (0.38)	1.17 (0.41)	0.569
Museums/visits	1.05 (0.21)	1.23 (0.48)	<0.001 *
Sports equipment	1.20 (0.60)	1.24 (0.59)	0.202
Others	1.27 (0.72)	1.51 (1.25)	0.090

Mann-Whitney U test *p*-values. * *p* is significant <0.05. M = mean value; SD = standard deviation. Each score obtained is based on a Likert scale (1–7): 1 “From 0 to 14 euros”, 2 “from 15 to 29 euros”, 3 “from 30 to 44 euros”, 4 “from 45 to 59 euros”, 5 “from 60 to 74 euros”, 6 “from 75 to 89 euros”, 7 “from 90 to 104 euros”.

**Table 5 ijerph-20-03889-t005:** Perception of the impact of kayaking in the Valle del Jerte.

Variable	Gender		*p*
	Female M (SD)	Male M (SD)	
Increased recognition and promotion of the destination	4.16 (0.98)	4.20 (0.89)	0.970
It represents an element of pride and satisfaction for the Valle del Jerte	3.78 (1.51)	3.43 (1.73)	0.075
Generates a traffic and safety problem	1.41 (0.49)	1.30 (0.52)	0.004 *
Disrupts the usual rhythm and impairs other activities	2.17 (1.01)	2.02 (1.17)	0.011 *
Increases in the number of overnight stays	3.28 (1.53)	3.24 (1.62)	0.800
Employment increases	3.70 (1.50)	3.35 (1.70)	0.023 *
It entails economic losses because the investment is greater than the benefits obtained	1.91 (0.99)	1.80 (1.01)	0.062
Economic advantages are concentrated in the hands of a few companies and individuals	2.49 (1.11)	2.24 (1.15)	0.005*
Public spending is necessary for the development of this type of sports	1.73 (0.92)	1.89 (1)	0.663

Mann-Whitney U test *p*-values. * *p* is significant <0.05. M = mean value; SD = standard deviation. Each score obtained is based on a Likert scale (1–5): 1 “Strongly disagree”, 2 “Disagree”, 3 “Indifferent”, 4 “Agree”, 5 “Strongly agree”.

**Table 6 ijerph-20-03889-t006:** Satisfaction with the contracted kayak service.

Variable	Gender		
**Would You Come Back to the Valle del Jerte to Do the Same or Other Sports Activities in Nature?**	**Female** **N (%)**	**Male** **N (%)**	** *p* **
Yes	288 (94.4)	197 (95.6)	0.543
No	17 (5.6)	9 (4.4)	
**Satisfaction with the Service Provided (Kayak) Has Been?**	**Female** **M (SD)**	**Male** **M (SD)**	** *p* **
	9.91 (1.31)	9.89 (1.60)	0.207

Chi-Square (item 1) and Mann-Whitney U test (item 2) *p*-values * *p* is significant <0.05. M = mean value; SD = standard deviation; N = number; % = percentage.

## Data Availability

The datasets are available through the corresponding author on reasonable request.

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
