# Peer review of "Gender Differences in Adventure Tourists Who Practice Kayaking in Extremadura"

_ijerph, 2023, doi:10.3390/ijerph20053889_

Round 1
Reviewer 1 Report
The presented article is interesting and has a proper structure for a research paper. However, we see that there are quite a few shortcomings.
It is not possible to agree with the title of the article, which does not have a scientific object. Kayaking in Valle del Jerte (Spain) is a practical activity promoting one area. Multidimensional analysis is a method that should be presented in the methodology section. If this is a case study, explain what makes the area special in the appropriate section.
The introduction is written in such a way that it is difficult to understand what it is trying to justify. Incredibly, the word Kayaking is mentioned here once and the word Adventure tourism is also mentioned once until Line 89. But they are the essential variables of the study.
None of the used questionnaires are validated. Apart from those related to socio-demographic questions and questions about how much money was spent. Validity is not relevant for them.
The results provide answers to the questions where the comparison between males and females is made. But the tasks of the research were much more than just assessing gender.
The conclusions lack scientific justification and are suitable for practical and economic assessment of the situation.
Author Response
It is not possible to agree with the title of the article, which does not have a scientific object. Kayaking in Valle del Jerte (Spain) is a practical activity promoting one area. Multidimensional analysis is a method that should be presented in the methodology section. If this is a case study, explain what makes the area special in the appropriate section.
Response: the title of the manuscript has been modified to take your recommendations into account.
The introduction is written in such a way that it is difficult to understand what it is trying to justify. Incredibly, the word Kayaking is mentioned here once and the word Adventure tourism is also mentioned once until Line 89. But they are the essential variables of the study.
Response: the introduction has been modified following your recommendations and those of other reviewers. It has been restructured and rewritten. In addition, a literature review section has been added to help the reader understand the different variables included in the analysis.
None of the used questionnaires are validated. Apart from those related to socio-demographic questions and questions about how much money was spent. Validity is not relevant for them.
Response: We are currently drafting the validation article of the different scales that focus their analysis on expenditure and the perception of the impact on the activity. Initially, it was planned to include them in this study but the sample was not sufficient to carry out a methodologically correct validation, so the sample is being expanded to include other adventure activities.
The results provide answers to the questions where the comparison between males and females is made. But the tasks of the research were much more than just assessing gender.
Response: This study aims to analyze the differences according to gender in the different variables evaluated by means of the scales and questionnaires. It is true that in the introductory section this was not very clear, so the last paragraph of the introduction (which contains the objective) has been rewritten.
The conclusions lack scientific justification and are suitable for practical and economic assessment of the situation.
Response: The conclusions show the most relevant findings of the study presented. Following your recommendations, references have been included in this section to support our findings.
Reviewer 2 Report
Dear author/s,
the topic of the manuscript is interesting, however there should be underline the originality of the manuscript and the contribution to the filed and the exiting literature.
The statistics is quite basic, I recommend the authors to improve this part (e.g. PCA).
Please mention the managerial implications, limitations and future research directions.
Good luck!
Author Response
the topic of the manuscript is interesting, however there should be underline the originality of the manuscript and the contribution to the filed and the exiting literature.
Response: This issue has been mentioned in the introduction and discussion following your recommendations.
The statistics is quite basic, I recommend the authors to improve this part (e.g. PCA).
Response: Please, understand that we are currently drafting the validation article of the different scales that focus their analysis on expenditure and the perception of the impact on the activity. Initially, it was planned to include them in this study but the sample was not sufficient to carry out a methodologically correct validation, so the sample is being expanded to include other adventure activities. As this is the first research in Spain that analyzes gender differences in the context of adventure tourism, we consider that although, as you say, it is a simple analysis, it is fundamental for the development of this research line.
Please mention the managerial implications, limitations and future research directions.
Response: these subsections have been included following your recommendations.
Good luck!
Reviewer 3 Report
As a whole the article is interesting and acceptable. But I think you have to take care of two aspects:
1. That the abstract coincides with the topic of the monograph in question. It is not the profile of the tourist that is important, but the issues of gender, age...
2. The conclusions are few, they respond more to a summary than to some conclusions (correlation), and they must respond to the set of topics addressed, but corresponding to the topics of the monograph.
Author Response
- That the abstract coincides with the topic of the monograph in question. It is not the profile of the tourist that is important, but the issues of gender, age...
Response: We understand that this issue had not been clarified as you indicated. The objective of the study has been reworded both in the abstract and in the introduction to facilitate the reader's understanding following your recommendation.
- The conclusions are few, they respond more to a summary than to some conclusions (correlation), and they must respond to the set of topics addressed, but corresponding to the topics of the monograph.
Response: The conclusions have been rewritten following your recommendation and including the issues you mentioned.
Reviewer 4 Report
Dear authors, thank you for giving me the opportunity to read your manuscript.
The article presents an interesting topic. The overview is clear and well presented. I found that, in the current context, the topic addressed in the paper is certainly important.
· I think that a chapter related to "Literature Review" would succeed in linking the general information with the specific ones analyzed.
· The conclusions of the study should address the limitations and opportunities of the study for future research much more clearly so as to provide some guidance on what should follow from their research and findings.
Author Response
I think that a chapter related to "Literature Review" would succeed in linking the general information with the specific ones analyzed.
Response: This section has been included following your recommendation. Thank you very much for the same, since after rereading the manuscript it was difficult to follow as it dealt with different aspects to take into account when characterizing the tourist. We hope that it has been simplified and structured in a convenient way.
- The conclusions of the study should address the limitations and opportunities of the study for future research much more clearly so as to provide some guidance on what should follow from their research and findings.
Response: This subsection has been included following your recommendation.
Round 2
Reviewer 1 Report
Comments for editor only